# Role of Angiogenesis in the Pathogenesis of NAFLD

**DOI:** 10.3390/jcm10071338

**Published:** 2021-03-24

**Authors:** Lin Lei, Haquima EI Mourabit, Chantal Housset, Axelle Cadoret, Sara Lemoinne

**Affiliations:** 1Centre de Recherche Saint-Antoine (CRSA) and Institute of Cardiometabolism and Nutrition (ICAN), Sorbonne Université, INSERM, 75012 Paris, France; lin.lei@inserm.fr (L.L.); haquima.el-mourabit@inserm.fr (H.E.M.); chantal.housset@inserm.fr (C.H.); axelle.cadoret@inserm.fr (A.C.); 2Department of Hepatology, Reference Center for Inflammatory Biliary Diseases and Autoimmune Hepatitis (CRMR, MIVB-H), Assistance Publique-Hôpitaux de Paris (AP-HP), Saint-Antoine Hospital, 75012 Paris, France

**Keywords:** non-alcoholic steatohepatitis, liver sinusoidal endothelial cells, vascular endothelial growth factor

## Abstract

Non-alcoholic fatty liver disease (NAFLD) has become the leading cause of chronic liver disease, exposing to the risk of liver fibrosis, cirrhosis, and hepatocellular carcinoma (HCC). Angio-genesis is a complex process leading to the development of new vessels from pre-existing vessels. Angiogenesis is triggered by hypoxia and inflammation and is driven by the action of proangiogenic cytokines, mainly vascular endothelial growth factor (VEGF). In this review, we focus on liver angiogenesis associated with NAFLD and analyze the evidence of liver angiogenesis in animal models of NAFLD and in NAFLD patients. We also report the data explaining the role of angiogenesis in the progression of NAFLD and discuss the potential of targeting angiogenesis, notably VEGF, to treat NAFLD.

## 1. Introduction

Non-alcoholic fatty liver disease (NAFLD) has become the most common cause of chronic liver disease worldwide [1]. NAFLD includes non-alcoholic fatty liver (NAFL) defined by steatosis and non-alcoholic steatohepatitis (NASH) defined by steatosis associated with inflammation and hepatocyte ballooning. NASH can progress to fibrosis, cirrhosis and hepatocellular carcinoma (HCC), accounting for a significant health burden. However, in spite of its high prevalence and significant health burden, there is currently no approved pharmacological therapy to treat patients with NAFLD.

Angiogenesis is a complex process leading to the development of new vessels from pre-existing vessels. Angiogenesis occurs under physiological conditions during normal wound healing and also in pathological contexts, such as tumorigenesis, so that antiangiogenic molecules (e.g., of Bevacizumab, an anti-vascular endothelial growth factor (VEGF) monoclonal antibody) are used in the treatment of different cancers, including HCC, according to recent guidelines [2]. Molecular and cellular mechanisms of angiogenesis have been extensively studied and are explained in detail elsewhere [3]. In short, in the quiescent state, endothelial cells are organized in a monolayer of cells, connected by intercellular junctions and surrounded by pericytes, which control their proliferation via the secretion of survival signals, such as VEGF and angiopoietin-1. When a vessel receives a proangiogenic signal such as VEGF, angiopoietin-2 (Ang-2), fibroblast growth factor (FGF) or another chemokine secreted by a hypoxic cell or an inflammatory cell, first, peri-cytes detach from the vessel in response to Ang-2. Then, endothelial cells become activated, lose their intercellular junctions and proliferate. VEGF increases the endothelial permeability, leading to the extravasation of plasma proteins which will make the scaffold of a temporary extracellular matrix. Endothelial cells migrate on the surface of this new extracellular matrix, forming a stalk that progressively elongates to build a new vessel. Once the neovessel is built, a maturation step is required to make the vessel functional. During this maturation step, endothelial cells go back to a quiescent state and pericytes are recruited to surround the endothelial cells, under the action of platelet-derived growth factor (PDGF), angiopoietin-1, transforming growth factor-ß (TGF-ß) and Notch. VEGF, the main proangiogenic cytokine, plays a central role in angiogenesis since it is implicated in all steps of angiogenesis: VEGF increases vascular permeability, induces endothelial cell proliferation and regulates neovessel lumen diameter [3]. Angiogenic effects of VEGF are mediated via its receptor VEGF receptor-2 (VEGFR-2), which is a tyrosine kinase receptor, expressed at the surface of endothelial cells [3].

Angiogenesis also takes place during chronic liver diseases. Indeed, liver fibrosis progression is accompanied by angiogenesis, regardless of the etiology of the liver disease [4,5]. In this setting, angiogenesis is triggered by hypoxia and inflammation, and its main effect is the aggravation of liver fibrosis, leading to cirrhosis [6].

In the context of chronic liver disease, angiogenesis leads to quantitative changes of liver vessels with the emergence of new vessels but also consists in qualitative changes of vessels (both pre-existing and new vessels), resulting in a process known as vascular remodeling. Such qualitative vascular changes include dedifferentiation of liver sinusoidal endothelial cells (LSECs), also called capillarization, defined by the loss of their fenestrae and the acquisition of a basement membrane [7]. In the studies of angiogenesis during chronic liver disease, it is particularly difficult to discriminate LSECs from vascular endothelial cells in the liver, first because not a single marker is fully specific of LSECs, and also because LSECs lose the expression of their canonical markers when they undergo capillarization [7].

This review focuses on angiogenesis associated with NAFLD. First, we report the manifestations of angiogenesis in animal models and in patients with NAFLD. Second, we address the role of angiogenesis in the progression of NAFLD. Finally, we discuss the potential of targeting angiogenesis to treat patients with NAFLD.

## 2. Evidence of Angiogenesis in NAFLD

Liver angiogenesis can be assessed either directly, typically by showing an increase in the number of hepatic vessels, or indirectly, by measuring the expression of angiogenesis markers, such as markers of endothelial cells or proangiogenic cytokines. Because indirect methods are the most easily accessible, they are commonly used, but one should keep in mind their limitations. Indeed, the expression of endothelial cell markers does not necessarily correlate with the number of endothelial cells because some endothelial markers, such as CD105, are upregulated during the activation of endothelial cells associated with angiogenesis [8].

### 2.1. In Animal Models

Many rodent models have been used to study NAFLD: each of them mimicking one or several features of human NAFLD, i.e., steatosis, NASH, fibrosis and HCC [9]. Several studies have reported the manifestations of angiogenesis in these models (Table 1). First, it was shown that the expression of CD31, the most commonly used marker of endothelial cells, was increased in the liver of mice fed with a high-fat diet (HFD) along with an increase of VEGFR-2 expression [10]. In rats fed with a choline-deficient amino acid (CDAA) diet, it was also shown that the expression of CD31 was increased in correlation with the stage of fibrosis [11]. The same result was observed with von Willebrandt factor (vWF), another classical marker of endothelial cell, in mice fed with a methionine- and choline-deficient (MCD) diet [12]. Others reported the induction of CD105 expression, a marker of activated endothelial cells which have acquired a proangiogenic phenotype, in the LSECs of mice fed with an MCD diet [12]. An increase in VEGF protein was reported in the liver of rats fed with a CDAA diet [11] and of mice fed with an MCD diet [12]. Interestingly, in the latter model, VEGF mRNA levels were unchanged, suggesting mechanisms of post-transcriptional regulation. Angiopoietin-2 (Ang-2) was increased in both the serum and livers of mice fed with an MCD diet or mice fed with streptozotocin (STZ) combined with a Western diet [13].

Specific techniques of imaging were used to analyze the global vasculature of mouse liver with NAFLD. Using scanned electronic microscopy of vascular corrosion casts of the liver, Coulon et al. showed that NAFLD was associated with a global alteration of the hepatic vascular architecture that consisted not only in an increased number of vessels but also in a clearly different phenotype of vessels, which displayed an enlarged diameter and a disrupted organization [12]. The same result was observed by Lefere et al. using micro-computed tomography [13], a technique that has the advantage of exploring the vasculature of the whole body and also in a live animal.

### 2.2. In Patients

Compared to studies in animal models, data related to liver angiogenesis in patients with NAFLD are more limited. We showed that the liver from patients with NAFLD displayed an increased expression of the endothelial marker, vWF especially in patients with advanced fibrosis [14]. Furthermore, in livers of NAFLD patients, we observed a correlation between the expression of vWF and the expression of collagen XV, a specific marker of portal myofibroblasts, that have proangiogenic properties, notably by secreting VEGF-containing extracellular vesicles [14]. Using immunostaining of CD34, a marker of neovascularization, Kitade et al. showed the presence of neovessels in livers from NASH patients, whereas these neovessels were absent in simply steatotic and normal livers [15]. Interestingly, in this study, the microvessel density was found to be proportional to the stage of liver fibrosis, cirrhotic livers showing the highest microvessel density [15]. These results were confirmed by Lefere et al. who showed that NAFLD patients displayed significantly increased CD34+ neovessels in their livers compared to healthy patients and patients with simple steatosis. Besides, the quantity of CD34 staining was shown correlated with Ang-2 serum levels [13].

NAFLD patients display increased serum levels of angiogenic markers such as VEGF, soluble VEGFR-1 (sVEGFR1) and sVEGFR2 [16]. However, a recent study did not confirm an increased level of VEGF, as opposed to vascular cell adhesion molecule-1 (VCAM-1), a marker of endothelial activation, in the serum of NAFLD patients [17]. The increased expression of proangiogenic cytokines was also reported in the livers of NAFLD patients [16]. Interestingly, the expression of VEGF and VEGFR1 mRNA was higher in livers with pure steatosis than in livers with NASH, suggesting an early induction of angiogenesis during the spectrum of NAFLD [16]. More recently, patients with biopsy-proven NASH were shown to have significantly higher serum levels of Ang-2 than those without NAFLD or those with simple steatosis [13]. In human steatosis and NASH, hepatic content in Ang-2 (protein) is also increased [13].

## 3. Role of Angiogenesis in the Progression of NAFLD

Evidence of liver angiogenesis in animal models of NAFLD and in NAFLD patients suggests a role of liver angiogenesis in NAFLD pathogenesis. During NAFLD, angiogenesis drives inflammation and fibrosis, as reviewed elsewhere [18]. Liver angiogenesis is not a specific event in NAFLD as it occurs in all chronic liver diseases with the progression of liver fibrosis [4,5,14]. However, specific features of angiogenesis have been reported in NAFLD. First, the upregulation of proangiogenic genes occurs very early during NAFLD progression, at the stage of pure steatosis, before NASH development, suggesting that angiogenesis is an early event during the physiopathology of NAFLD [12]. Steatosis by itself is able to induce hypoxia, which is the primary inducer of VEGF and trigger of angiogenesis. Indeed, steatosis by boosting the metabolism of fatty acids increases oxygen consumption, generating a hypoxic proangiogenic micro-environment. Steatosis also causes mechanical pressure on sinusoids [19], which further aggravates the shortage of oxygen supply, explaining why patients with pure steatosis may develop portal hypertension [20]. These two mechanisms probably lead to hypoxia and explain why steatosis by itself induces the expression of VEGF [21]. Moreover, hypoxia could be more critical in NAFLD compared to other chronic liver diseases since injury in NAFLD primarily occurs in the perivenular zone, which is more susceptible to hypoxia than the periportal zone, that is primarily injured in other types of liver diseases such as viral hepatitis or biliary diseases. Finally, liver angiogenesis mainly involves LSECs [12], which play a key role in NAFLD pathogenesis, as reviewed elsewhere [22]. For all these reasons, one may infer that angiogenesis is particularly intense in NAFLD, although a comparison of angiogenesis in NAFLD versus other chronic liver diseases, either in animal models or in patients, is lacking.

The molecular pathways of angiogenesis are intermingled with those of NAFLD. Proangiogenic cytokines have an impact on NAFLD. Indeed, VEGF is also involved in lipogenesis so that anti-VEGFR2 treatment can induce changes in the expression of lipogenesis genes, as shown in MCD mice [12]. Hypoxia-inducible factors (HIFs), master effectors of hypoxia, also regulate the expression of genes involved in glucose metabolism [23]. Conversely, cytokines involved in NAFLD development have an impact on angiogenesis. For instance, leptin, an adipokine, which regulates satiety with a key role in obesity, has been shown to stimulate angiogenesis [24]. Kitade et al. showed that leptin-deficient rats exposed to a CDAA diet developed NASH but without neovascularization, as opposed to wild-type rats, clearly indicating that leptin is necessary for angiogenesis in NAFLD [11]. Another example is TGF-β, a master profibrogenic cytokine, which promotes the activation of liver myofibroblasts [25]. TGF-β has also proangiogenic capacities, notably by positively regulating the pericyte differentiation, proliferation and migration [26]. TGF-β is upregulated in NAFLD patients [27] and is increased in patients with NASH compared to patients with pure steatosis [28]. Angiotensin II is known to have profibrogenic properties [29] and angiotensin II-mediated signaling is involved at multiple levels in the development and progression of NAFLD [30]. Angiotensin II has also proangiogenic capacities. Particularly, angiotensin II induced the expression of VEGF in hepatic stellate cells [31].

Increasing evidence indicates that macrophages play a critical role in NAFLD development and progression [32,33]. Yet, macrophages also exert proangiogenic properties in chronic liver diseases. RNA sequencing analysis showed that macrophages isolated from the liver of MCD mice (especially monocyte-derived macrophages) expressed not only inflammatory cytokines but also growth factors involved in angiogenesis [34]. Furthermore, Miura et al. demonstrated that the macrophages were the source of VEGF production in steatotic livers from MCD mice [21]. Yet, in non-NAFLD animal models of chronic liver disease, monocytes-derived macrophages also accumulate in injured livers and exhibit proangiogenic gene profiles, including upregulated VEGF expression [35]. In this latter study, the inhibition of monocyte infiltration prevented angiogenesis but not fibrosis progression, demonstrating the direct role of macrophages in angiogenesis and no strict dependence between angiogenesis on one hand and liver fibrosis on the other [35].

One of the mechanisms that promote angiogenesis is the secretion of extracellular vesicles. Extracellular vesicles are submicron membrane-bound structures secreted from different cell types. They contain a wide variety of molecules and exert important functions in cell-to-cell communication [36]. Many studies have reported the angiogenic pro-perties of extracellular vesicles in different settings [14,37]. Extracellular vesicles are also implicated in the pathophysiology of liver diseases [38,39]. Povero et al. showed that hepatocytes exposed to free fatty acids in vitro released extracellular vesicles able to induce angiogenesis, both in vitro and in vivo, via vanin-1-dependent mechanisms [40]. In this study, the authors observed high levels of circulating hepatocyte-derived extracellular vesicles that were associated with marked liver angiogenesis in mice fed with an MCD diet [40]. Mechanisms promoting angiogenesis in NAFLD are illustrated in Figure 1.

Cirrhosis is associated with the risk of developing HCC in all chronic liver diseases including NAFLD. However, non-cirrhotic patients with NASH have a higher risk of HCC compared to patients with other types of liver disease [41]. HCC is a highly vascularized tumor, a feature that is exploited in the imaging-based diagnosis of HCC. Angiogenesis is a key driver of HCC, and the tumoral expression of angiogenic factors, notably Ang-2, is associated with a pejorative prognosis in HCC [42]. As mentioned earlier, Ang-2 is overexpressed in the liver of patients with NAFLD [13]. Therefore, more sustained angiogenesis could contribute to a higher risk of developing HCC in NAFLD.

Ultimately, the best way to demonstrate the role of angiogenesis in the pathogenesis of NAFLD is to assess the effect of antiangiogenic treatments in NAFLD.

## 4. Angiogenesis as a Potential Therapeutic Target in NAFLD

A few antiangiogenic molecules have been tested in animal models of NAFLD. Coulon et al. showed that treatment with anti-VEGFR2 decreased steatosis and inflammation in MCD mice diet [12] and reduced the disruption of the liver vasculature even though the vasculature was not normalized, either in preventive or therapeutic settings [12]. Interestingly, in the NASH model they used, liver fibrosis is of low intensity (stage 1, Table 1) and the authors observed no effect of anti-VEGFR2 on liver fibrosis. Therefore, targeting VEGF seems efficient to improve NASH in animal models. Nevertheless, the VEGF signaling pathway is not restricted to angiogenesis. Indeed, VEGF also plays a direct role in fibrogenesis since VEGF increases fibrogenic functions of hepatic stellate cells, such as collagen I secretion [43] and migration [44]. Moreover, as mentioned earlier, VEGF is also involved in lipogenesis, and treatment with anti-VEGFR2 significantly decreased lipid accumulation in fat-laden primary hepatocytes in vitro [12]. Hence, interventions that target VEGF or its receptor do not act only on angiogenesis.

Placental growth factor (PlGF) is a proangiogenic cytokine associated selectively with pathological angiogenesis. A previous study has shown that inhibition of PlGF reduced angiogenesis, inflammation and fibrosis in a non-NAFLD animal model of liver disease [45], whereas Coulon et al. observed no effect of treatment with anti-PlGF on NASH in MCD mice [12]

Besides anti-VEGFR2, another antiangiogenic agent has been assessed in NASH, i.e., the peptibody L1-10 which inhibits the interaction of Ang-2 with its receptor Tie2 [13]. Administration of L1-10 in mice fed with an MCD diet decreased the hepatic vascular density and partially corrected the disorganization of the vascular network, as demonstrated by the compelling images of scanning electron microscopy [13]. Moreover, administration of L1-10 reduced the severity of hepatocellular ballooning and fibrosis (without any effect on steatosis) in two different mouse models of NASH: MCD diet and STZ–Western diet [13], the latter model displaying a higher stage of liver fibrosis than the first one (Table 1). Of note, this antiangiogenic treatment did prevent HCC progression in the STZ–Western model [13].

The inhibition of formation of extracellular vesicles (by genetic inhibition of caspase 3) has been shown to limit the production of hepatocyte-derived proangiogenic extracellular vesicles and to protect mice fed with MCD diet from angiogenesis and fibrosis, independent of steatosis and inflammation [40].

The treatment of CDAA rats with angiotensin II type 1 receptor blocker has been shown to decrease neovascularization and liver fibrosis [31]. These results were also confirmed by Tamaki et al., who showed that in addition to decreased angiogenesis and fibrosis, angiotensin II type 1 receptor blocker also reduced HCC development in CDAA rats [46]. Effects of antiangiogenic molecules on NAFLD are illustrated in Figure 1.

One should note that the antifibrotic effects of the antiangiogenic treatments are not specific to NAFLD and have also been reported in other models of chronic liver diseases [35,45,47,48,49,50,51,52,53,54]. In addition, the effect of antiangiogenic treatment on liver fibrosis can be different according to the time point of agent administration. Indeed, during the resolution of fibrosis, angiogenesis could be helpful. Inhibition of VEGF by neutralizing antibody decreases liver fibrosis induced by bile duct ligation in rats, whereas it prevents regression of fibrosis after the biliary obstacle is removed [54].

In conclusion, many studies have provided convincing evidence of early and intensive liver angiogenesis in NAFLD pathogenesis both in animal models and in patients. Angiogenesis promotes the development of NAFLD, the progression of fibrosis and the emergence of HCC. Antiangiogenic treatment can reduce NASH and prevent HCC formation in different animal models. Antiangiogenic molecules approved to treat advanced hepatocellular carcinoma, notably Bevacizumab [2], could have a beneficial impact on NAFLD.

## Figures and Tables

**Figure 1 jcm-10-01338-f001:**
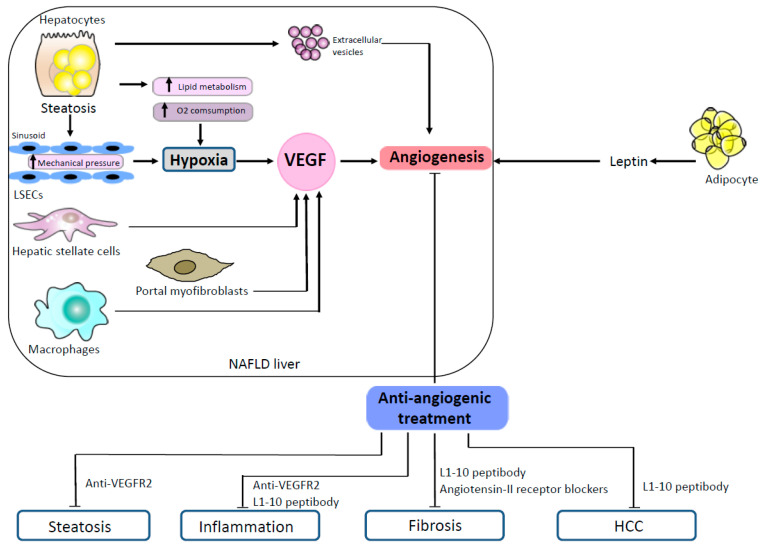
Mechanisms promoting angiogenesis in NAFLD and effects of antiangiogenic treatments in animal models. In NAFLD, steatotic hepatocytes produce proangiogenic extracellular vesicles. Steatosis induces hypoxia both by an increased lipid metabolism which enhances oxygen consumption and by a mechanical pressure on sinusoids. Hepatic stellate cells, portal myofibroblasts and macrophages stimulate angiogenesis by secreting VEGF. Proangiogenic signals also come from the adipose tissue which secretes leptin. In animal models of NAFLD, several antiangiogenic treatments (anti-VEGR2, L1-10 peptibody, angiotensin II receptor blockers) have shown efficacy to reduce steatosis, inflammation, fibrosis and HCC. Abbreviations: HCC, hepatocellular carcinoma; LSECs, liver sinusoidal endothelial cells; NAFLD, non-alcoholic fatty liver disease; VEGF, vascular endothelial growth factor; VEGFR-2, vascular endothelial growth factor receptor-2.

**Table 1 jcm-10-01338-t001:** Study of angiogenesis in animal models of non-alcoholic fatty liver disease (NAFLD).

Model	Steatosis	NASH	Fibrosis	HCC	Description of Liver Angiogenesis	Reference
HFD mouse	+++	+	+	absent	Increase of CD31, VEGFR-2	[10]
MCD mouse	+++	+++	+	absent	Increase of vWF, CD105, VEGF, Ang-2 Increase of vascular density	[12]
CDAA rat	+++	+++	+++	present	Increase of CD31, VEGF	[11]
STZ + Western diet mouse	+++	+++	++	present	Increase of Ang-2 Increase of vascular density	[13]

Abbreviations: Ang-2, angiopoietin-2; CDAA, choline-deficient amino acid; HCC, hepatocellular carcinoma; HFD, high-fat diet; MCD, methionine- and choline-deficient diet; NASH, non-alcoholic steatohepatitis; STZ, streptozotocin; VEGF, vascular endothelial growth factor; vWF, von Willebrand factor. + mild, ++ moderate, +++ important.

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
