# Peer review of "Role of Angiogenesis in the Pathogenesis of NAFLD"

_jcm, 2021, doi:10.3390/jcm10071338_

Round 1

Reviewer 1 Report

I also think that angiogenesis is important for the pathogenesis of non-alcoholic fatty liver disease (NAFLD). This article is difficult to understand for lack of figure and table. The other article "Angiogenesis in the progression of non-alcoholic fatty liver disease" (Acta Gastro-Enterologica Belgica, Vol. LXXXIII, April-June 2020) is very easy to understand. Because these articles are very similar, the author should revise this article referencing the other articles for publication. 

Author Response

  • As suggested, we included 1 figure and 1 table.
  • We also added the publication from Lefere et al., Acta Gastroenterol Belg, 2020 in our references (cited page 6).

Reviewer 2 Report

This is a review on the current data on angiogenesis in NAFLD and angiogenesis as a potential therapeutic target. The scientific evidence, unfortunately, in this field is limited mainly to animal models, whereas clinical research is scarce. The review summarises this limited evidence focusing in the following three domains:

Evidence of angiogenesis in NAFLD

  • The authors report on levels of different markers of angiogenesis in animal models of NAFLD and patients with NAFLD. They do not go into much detail and depth. It would be useful to mention the role of these different markers in the process of angiogenesis.

Role of angiogenesis in the progression of NAFLD

  • My understanding is that the authors are suggesting an interplay between metabolic pathways in NAFLD and angiogenesis. They describe potential mechanisms/pathways, but again without going into much depth. A figure with all the proposed mechanisms might be useful and informative.

Target angiogenesis to treat NAFLD

  • Angiotensin II is only mentioned in this section. Since angiotensin II plays a role in angiogenesis, it should be also mentioned in the previous section.
  • Are there any data on the effect of anti-angiogenic therapy, such as Bevacizumab, in patients with NAFLD?

Author Response

  • In the introduction, we added a paragraph explaining the main steps of angiogenesis and the role of the different proangiogenic cytokines.
  • As suggested, we also provide a figure illustrating the mechanisms of angiogenesis in NAFLD.
  • We added a comment regarding the angiogenic role of angiotensin II (page 7).
  • Unfortunately, there is no data on the effect of bevacizumab on NAFLD. The studies, which evaluated the effect of bevacizumab on colorectal cancer (Nalbantoglu J Clin Dise, 2014) or HCC (Finn, N Engl J Med, 2020) have included patients with NAFLD but none of these studies has specifically addressed the effect of bevacizumab on NAFLD. Since Bevacizumab has been recently approved as a first line therapy for unresectable HCC (Finn, N Engl J Med, 2020), we should have more data in a near future.

Reviewer 3 Report

The authors reviewed the role of angiogenesis in the pathogenesis of NAFLD. It is an important topic to understand the progression of NAFLD and to investigate the therapeutic targets for NAFLD. This review article is well-written and covers the angiogenesis topic. However, some issues should be addressed.

  1. There is no information of TGF-b, which is also a key player for angiogenesis. Authors should describe how TGF-b involves in the angiogenesis and also includes the connection between angiogenesis and fibrosis.
  2. Some citated review papers, not original papers, are old. Authors should check new review papers if they can replace.

Author Response

  • As suggested, we added a section about the role of TGF-beta in angiogenesis (page 7).
  • We followed your recommendation and replaced some cited reviews by more recent ones (new references 5, 6, 8, 23, 24, 25).

Round 2

Reviewer 1 Report

With the addition of table and figure, the review article has become much easier to understand.

If the author considers the following five points, I think it acceptable.

①The author needs to modify the italic "fenestrae" to the roman "fenestrae".

②The author must provide a citation on “Indeed, the expression of endothelial cell markers do not necessarily correlate with the number of endothelial cells because some markers, such as CD34, are upregulated during the activation of endothelial cells associated with angiogenesis”.

③Coulon et al. are not the authors of reference 12, so the authors need to remove 12 from "organization 11,12".

④The author needs to remove the underline of “Role of angiogenesis in the progression of NAFLD”.

⑤Bevacizumab is not the only angiogenesis inhibitor used to treat liver cancer, but Sorafenib, Regorafenib, Lenvatinib, Ramucirmab, and Cabozantinib are also available, so it is desirable to exclude the last sentence of the discussion.

Author Response

We thank the reviewer for the useful corrections.

1) We did replace “fenestrae” by “fenestrae”.

2) CD34 is classically known as a marker of neovessels, especially in cancer. However, we did not find any original article demonstrating that this marker is upregulated in endothelial cells during angiogenesis. So, in this sentence, we replaced CD34 with CD105 with an appropriate reference.

3) It was our mistake. We removed the inappropriate reference at the end of the sentence starting with "Coulon et al."

4) The underline was removed.

5) We totally agree that like Bevacizumab, Sorafenib, Regorafenib, Lenvatinib, Ramucirmab and Cabozantinib are also antiangiogenic molecules used to treat liver cancer, but contrary to Bevacizumab, these molecules have not been shown to improve NASH in animals models. We modified the last sentence of the manuscript.